# A broader lens on tuberculosis cost-effectiveness analysis: How patient-incurred costs and post-tuberculosis outcomes reshape estimates in a multi-country study

Ewan M. Tomeny[1]*, Phuong Bich Tran[2], Joseph Kazibwe[3], Laura Rosu[1], Georgios F. Nikolaidis[4], Rebecca Nightingale[1,5], Tom Wingfield[1,6,7], Jamilah Meghji[8], S. Bertel Squire[1,7], Eve Worrall[1]

1 Centre for Tuberculosis Research, Departments of Clinical Sciences and International Public Health, Liverpool School of Tropical Medicine, Liverpool, United Kingdom, 2 Nuffield Department of Primary Care Health Sciences, University of Oxford, Oxford, United Kingdom, 3 Department of Clinical Sciences, Lund University, Malmö, Sweden, 4 Methods and Evidence Generation Department, Centre of Excellence in Evidence Synthesis, IQVIA Ltd, London, United Kingdom, 5 Respiratory Department, Liverpool University Hospitals NHS Foundation Trust, Liverpool, United Kingdom, 6 Department of Global Public Health, Karolinska Institutet, Stockholm, Sweden, 7 Tropical and Infectious Diseases Unit, Liverpool University Hospitals NHS Foundation Trust, Liverpool, United Kingdom, 8 National Heart and Lung Institute, Imperial College London, London, United Kingdom

* ewan.tomeny@lstmed.ac.uk

**Data availability statement:** All data inputs used in this study are publicly available and are

## Abstract

Tuberculosis (TB) remains a major public health challenge, with financial and health impacts extending beyond treatment. Both the perspective adopted in cost-effectiveness analyses—which critically determines choices such as the inclusion of patient-incurred costs—and the extent to which long-term post-treatment considerations are incorporated have important policy implications. This study examines how the choice of timeframe and cost perspectives influence the estimated cost-effectiveness of TB interventions, particularly preventative measures. Using data from 19 WHO TB patient cost surveys and global epidemiological databases, we modelled a hypothetical preventative TB intervention, generating four incremental cost-effectiveness ratios (ICERs) per country under different analytical approaches. These included a conventional timeframe up to treatment completion, an extended timeframe incorporating post-TB effects, and two perspectives that either included or excluded patient-incurred costs. The approach yielding the lowest ICER (societal perspective; extended timeframe) was anchored in the primary analysis to a 1×GDP per capita threshold. Using this benchmark simplified cross-country comparisons and removed the need for health system cost estimates. Sensitivity and scenario analyses explored how threshold values influenced the relative impact of timeframe and costing perspective. ICERs were higher when patient costs were omitted or the post-TB period was excluded, peaking when both were absent. However, across all

cited in the manuscript. These, along with many intermediate values and outputs, are presented in the manuscript and appendices. The authors are happy to share any additional analytical materials upon request.

**Funding:** No authors received specific funding for this work.

**Competing interests:** The authors have declared that no competing interests exist.

countries, post-TB considerations had a far greater impact on cost-effectiveness. On average, removing the post-TB period increased ICERs by over 50% (ranging from +19.3% in Ghana to +108% in Mongolia, societal perspective). Including patient-incurred costs increased the likelihood that prevention was cost-effective, particularly in low-GDP settings with lower willingness-to-pay thresholds. However, their impact was minimal above 2×GDP. Our study highlights how narrowly defining the financial and health burden of tuberculosis in cost-effectiveness analyses risks underestimating the benefits of interventions—particularly in lower-GDP countries where the socioeconomic burden of tuberculosis is greatest—which could lead to misguided policy decisions that overlook the full impact of tuberculosis.

## Introduction

Tuberculosis (TB) significantly affects people not only through mortality but also via its impact on health, including functional capacity and psycho-social wellbeing [1,2]. The journey taken by someone with TB, from symptoms, through diagnosis and treatment to eventual recovery, often imposes a significant financial burden on patients and their households, including direct medical (e.g., diagnostic tests) and non-medical (e.g., transport) out-of-pocket expenses, alongside indirect costs, e.g., through loss of productivity due to illness [3]. The World Health Organization (WHO) has in the past decade highlighted the importance of patient-incurred costs, and—in part to assess progress toward the END-TB targets of eliminating 'catastrophic costs', [4]—many countries have now completed a National TB Patient Cost Survey. Findings summarised in 2022 reveal that "none of the surveyed countries [are] coming close to achieving the End TB strategy target of zero" (the 'catastrophic cost' milestone target for 2020) [3]. Furthermore, the WHO cost survey centres on costs of the care-seeking and treatment periods, with their handbook acknowledging that, while long-term effects of TB "can impair household economics for years", capturing these costs would require "a longer-term time-window than the present study design allows" [5]. While currently few, those studies that have investigated the long-term post-treatment economic impact of TB demonstrate a common finding: that the economic repercussions of TB are both considerable and destabilising [6–8].

The analytical perspective taken in a cost-effectiveness analysis determines which costs are included. Defining the perspective is, in essence, a normative question regarding social value, aligned with the decision maker upon whom the costs of implementing the intervention will fall. While agreement is lacking on how different perspectives are defined and overlap between definitions is common [9,10], taking a 'provider' or 'healthcare sector' perspective is more common than taking the broader 'societal' perspective (a review of over 7,000 cost-effectiveness analyses (CEAs) found 74% took one of these two narrower perspectives) [11]. Grounded in welfare economics [12], the societal perspective aims to capture the full economic impact of a health intervention across all members of society by incorporating all relevant costs and consequences, irrespective of who incurs them, and to maximise overall societal benefit. Despite its conceptual breadth, in practice analysts often face challenges in applying the full societal perspective due to limited data on non-health sector costs, such as productivity losses or informal caregiving time. Nevertheless, inclusion of at least some patient-incurred costs—such as direct payments and lost income—is widely considered a minimum expectation when adopting this perspective.

In addition to patient-cost data, the WHO TB patient-cost survey collects information on coping mechanisms and broader household impacts, which vary considerably by country. Coping strategies—such as asset sales, borrowing, and dissaving—were reported by 42% of TB-affected households across 20 countries (pooled average; 95% CI: 33–50%). Other socioeconomic consequences included missed school days for children (4.2% of households; ranging from 0.31% in Indonesia to 21% in the Solomon Islands), divorce or separation (3%; 0.36% in the Philippines to 24% in Kenya), and employment loss (23% overall; 1.8% in Fiji to 46% in Mongolia) [3]. Unlike conventional measures of catastrophic health expenditure (CHE), which focus primarily on medical costs relative to household income, TB Patient Cost Surveys capture a broader range of non-medical costs and lost income, providing a more comprehensive assessment of the financial impact of TB on patients and their households, and a more holistic consideration of patient costs in CEAs conducted from a societal perspective. Despite the documented socioeconomic burden of TB on patients and households, only one in four TB CEAs include patient costs—mirroring the broader trend in cost-effectiveness analyses [11,13]. Moreover, those studies taking a societal perspective rarely go further than the addition (or subtraction) of patient costs during treatment (and at times care-seeking) to those of the health system, which—among other broader considerations—critically overlooks the enduring financial shock to the household in the post-TB period, beyond treatment completion.

In addition to decisions on whose costs and what consequences to include, decisions are also required on the timeframe for which these are considered. Recommendations advise that the "time horizon adopted in CEA should be long enough to capture all relevant future effects of a health care intervention" [14]. Recent years have witnessed a growing focus and body of evidence on post-tuberculosis lung disease (PTLD) and other long-term health effects of tuberculosis beyond microbiological cure [15–19].

Units are needed in CEA to express health outcomes. The widely used generic outcome measure, the *Disability-adjusted life year* (DALY), is designed to provide an objective overview of a health intervention's effects by combining the years of life lost due to premature death (mortality) and years of healthy life lost due to impaired health (morbidity) [20]. While the DALY is the predominant measure in TB cost-effectiveness analyses, its application varies, particularly in how it accommodates non-fatal health consequences [13]. Research has demonstrated that post-TB morbidity significantly contributes to the global TB burden [17,21], and although it has been included in some cost-effectiveness analyses [22], such inclusion is exceptional [13].

Despite the now well-established evidence base for both the long-term health effects of TB and the considerable patient-incurred costs before, during, and after treatment, analyses only occasionally consider patient costs before and during treatment, and rarely include costs or health effects beyond treatment. It is likely that omissions of these factors are unfavourably skewing the perceived benefits of interventions, although the extent of this misrepresentation has not been clearly established and remains to be systematically explored in a cross-country analysis. This study examines how extending the time period and incorporating patient costs would influence the cost-effectiveness analyses of a hypothetical preventative TB intervention, such as a vaccine, when compared to no intervention. We utilise patient-cost data for both drug-susceptible TB and multidrug-resistant TB (MDR-TB) from 19 countries that have conducted robust national surveys, along with epidemiological data from global databases and existing literature. Employing transparent assumptions relating to cost structures, health outcomes, and epidemiological data, our analysis seeks to elucidate the ranges and magnitudes by which cost-effectiveness ratios would vary by country when considering longer time periods and differing costing perspectives. Through examining variations in cost-effectiveness ratios

against different willingness-to-pay per DALY averted thresholds, our study aims to provide context-sensitive insights to support policy-making decisions.

## Materials and methods

### Overview

We evaluated a single hypothetical preventative TB intervention using different analytical approaches that varied in terms of the time period considered and costing perspective. Modelled at the population level but acting at the individual level, this assessment sought to understand and broadly quantify how considerations beyond those typically employed in TB cost-effectiveness analysis affect cost-effectiveness ratios (CERs). (Nb. Hereafter we refer to these as *'ICERs'* (incremental CERs) in alignment with common terminology, though we are technically discussing CERs.)

### Approaches

Specifically, we considered two TB phases: a '*Conventional* TB-phase' that included only costs and health effects in the period until treatment completion, and an '*Extended* TB-phase' that included the Conventional TB-phase plus the post-TB period. For each evaluation, we separately assessed the intervention from two perspectives: first, a health system perspective, including only incremental costs incurred by the health system ($\Delta C_{\text{hs}}$), and second, a 'limited societal' perspective [11], which additionally accounted for costs incurred by TB patient households ($\Delta C_{\text{pat}}$) (Fig 1).

Outcomes were measured in DALYs, reported as 'DALYs averted', comprising reductions in TB-related mortality (Years of Life Lost, $\Delta D_{\text{YLL}}$) and TB-related morbidity (Years Lived with Disability, $\Delta D_{\text{YLD}}$). We focused on the direct health effects of the intervention—specifically reductions in TB incidence, deaths, and disability—without modelling transmission dynamics or indirect (herd) protection effects. A detailed description of how these effects were implemented is provided in section "Health outcomes and intervention effect modelling". All costs are expressed in USD, adjusted to a 2022 baseline. We applied our methodology to 19 countries, all LMICs, chosen due to availability of patient-cost data collected under the same WHO protocol [5].

In all approaches, incremental YLD were calculated for active disease ($\Delta D_{\text{YLD,aTB}}$), assumed to apply to time spent in treatment ($t_{\text{Tx}}$) and pre-treatment ($t_{\text{preTx}}$) periods. We assumed a 6-month treatment regimen for drug-susceptible TB (DS-TB) and a 9-month regimen for multidrug-resistant TB (MDR-TB). The pre-treatment duration (i.e., time spent in care-seeking) was varied in sensitivity analyses.

**Conventional period, health system perspective ('Basic Evaluation').** The most basic evaluation conforms to the conventional time period under a health system perspective, including health system costs during active TB ($\Delta C_{\text{hs,aTB}}$), TB mortality ($\Delta D_{\text{YLL}}$) and disability during active disease ($\Delta D_{\text{YLD,aTB}}$). These terms remained constant in all subsequent approaches.

$$\text{ICER}_{\text{Std,hs}} = \frac{\Delta \text{Costs}}{\Delta \text{DALYs}} = \frac{\Delta C_{\text{hs,aTB}}}{\Delta D_{\text{YLL}} + \Delta D_{\text{YLD,aTB}}}$$

**Conventional period, societal perspective.** The evaluation using the conventional period under a societal perspective included patient costs incurred during care-seeking and treatment ($\Delta C_{\text{pat,aTB}}$).

$$\text{ICER}_{\text{Std,soc}} = \frac{\Delta C_{\text{hs,aTB}} + \Delta C_{\text{pat,aTB}}}{\Delta D_{\text{YLL}} + \Delta D_{\text{YLD,aTB}}}$$

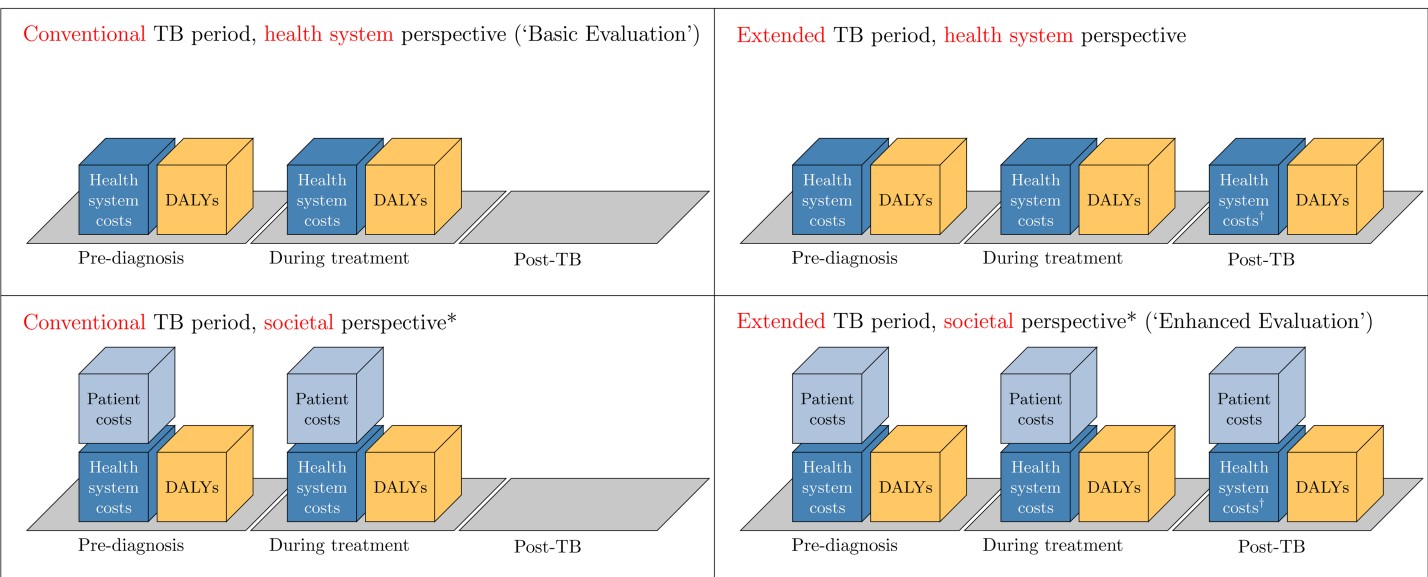

**Fig 1. Evaluation approaches and perspectives.** * In evaluating the societal perspective, our analyses are limited to additional costs borne by the patient's household. During the pre-diagnosis and treatment periods, these encompass direct and indirect costs as reported in WHO patient cost surveys. For the post-TB period, we assume a delayed financial recovery based on observed reductions in household finances at treatment completion. †Post-TB health system costs were set to zero in the core analyses due to a lack of available data but were varied in subsequent scenario analyses.

**Extended period, health system perspective.** The extended period included the long-term disability due to TB ($\Delta D_{\text{YLD,ltd}}$).

$$\text{ICER}_{\text{Extd,hs}} = \frac{\Delta C_{\text{hs,aTB}}}{\Delta D_{\text{YLL}} + \Delta D_{\text{YLD,aTB}} + \Delta D_{\text{YLD,ltd}}}$$

**Extended period, societal perspective ('Enhanced Evaluation').** The most comprehensive 'Enhanced Evaluation' considered the extended period under the societal perspective, including an additional cost term accounting for the financial shock on the patient's household post treatment completion ($\Delta C_{\text{pat,lts}}$).

$$\text{ICER}_{\text{Extd,soc}} = \frac{\Delta C_{\text{hs,aTB}} + \Delta C_{\text{pat,aTB}} + \Delta C_{\text{pat,lts}}}{\Delta D_{\text{YLL}} + \Delta D_{\text{YLD,aTB}} + \Delta D_{\text{YLD,ltd}}}$$

**Anchoring ICERs.** Assuming the reduction in TB prevalence from a preventative intervention would lead to a reduction in both TB patient costs and long-term disability, the 'Enhanced ICER' would be smallest (most favourable), the 'Basic ICER' the greatest, with the other two between them.

$$\text{i.e.,} \quad \Delta C_{\text{pat,aTB}}, \Delta C_{\text{pat,lts}} < 0 \implies \Delta C_{\text{soc}} < \Delta C_{\text{hs,aTB}},$$
$$\Delta D_{\text{YLD,ltd}} > 0 \implies \Delta D_{\text{Extd}} > \Delta D_{\text{Std}},$$
$$\implies \text{ICER}_{\text{Extd,soc}} < \text{ICER}_{\text{Std,hs}}$$

The four evaluation approaches differ only by the inclusion of the terms $\Delta C_{\text{pat,aTB}}$, $\Delta C_{\text{pat,lts}}$, and $\Delta D_{\text{YLD,ltd}}$. By adopting a simplifying heuristic, stipulating that the lowest ICER precisely meets the cost-effectiveness threshold ($\text{ICER}_{\text{Extd,soc}} = k$), we can determine the other ICERs

relative to this without requiring additional assumptions about the magnitude of incremental health system costs (Fig 2). While distinct, this approach shares similarities with CEA methodology where 'net benefits' are calculated utilising a predetermined cost-effectiveness threshold [23].

In simple terms, we set the most comprehensive analysis scenario (extended timeframe, societal perspective) to be exactly cost-effective at a threshold of 1×GDP per capita. This allowed us to derive all other ICERs relative to this benchmark without assuming an arbitrary intervention cost.

With our formulation using overall rates for input parameters (prevalence, population breakdowns, mortality, case-detection etc), it is inherently scale-invariant with respect to total population size, meaning no further assumptions are required regarding implementation costs.

Treatment-related disability ($\Delta D_{Tx}$) was not included in the core analysis, as we assumed no significant long-term adverse effects from the intervention. Scenario analyses exploring potential treatment-related disability are described in section "Adverse event from vaccination".

Post-TB health system costs—such as those related to managing long-term disability—were excluded from all approaches in the core analysis due to a lack of robust data across settings. However, their potential impact was explored through a separate scenario analysis (see section "Post-TB health system costs").

## Model parameters

**Population and epidemiological parameters.** TB incidence and mortality estimates for 2022 were obtained from WHO databases [24], which are provided disaggregated by sex and reported in broad age categories (e.g., 0–14, 15–24, 25-34) for each country. To convert these into more granular 5-year age groups, we used population distribution data from the Global Burden of Disease Study 2019 (IHME) [25], which provides sex-disaggregated estimates in 5-year age bands (e.g., 0–4, 5–9, etc.). Assuming that age-sex demographic structures have remained stable since 2019, we applied these proportions to disaggregate the WHO estimates.

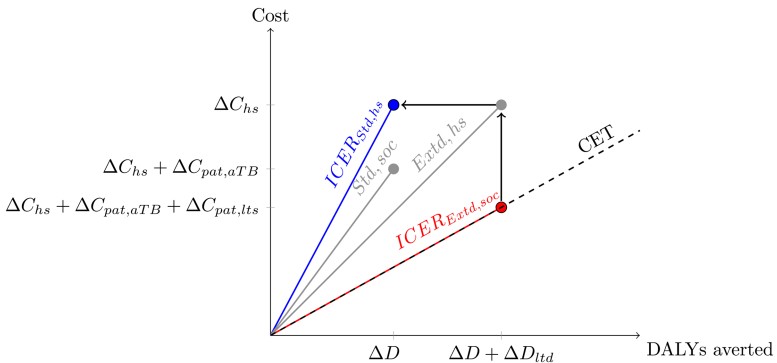

**Fig 2. Relating ICERs through the threshold.** Conceptual framework illustrating how ICERs transition between different analytical perspectives. The pink arrows indicate the transition from the Extended ICER under the societal perspective ('Enhanced ICER', red) to the ICER under the health system perspective calculated for the conventional TB period ('Basic ICER', blue) for a preventative TB intervention. This is depicted as a shift upward (representing an increase in total costs due to no longer considering the [negative] incremental patient costs under a HS perspective) and leftward (representing fewer DALYs averted through not including long-term disability). Assuming the Enhanced ICER falls on the cost-effectiveness threshold provides both HS costs and the Basic ICER.

This enabled us to estimate incidence and mortality for each 5-year sex-age subgroup, facilitating more detailed DALY calculations across demographic strata. HIV and MDR-TB prevalences among patients were additionally taken from the WHO [24]. Lacking detailed demographic breakdowns for MDR-TB prevalence, we assumed a uniform distribution across all age and sex groups as per the WHO overall country percentage estimate.

**Life expectancy.**  For each study country, age and sex-specific disaggregated estimates for life remaining (5-year age brackets) were obtained from the 2022 United Nations (UN) life tables [26]. These were used to YLLs, by assigning the remaining life expectancy to each averted TB death.

**TB treatment coverage.**  We utilised 2022 WHO case detection rates (CDR) for each country to estimate the proportions of active TB cases who would receive treatment. We assumed consistent rates for those with active TB both with and without the intervention.

**Health outcomes and intervention effect modelling.**  We modelled a generic preventive intervention (e.g., a vaccine) applied uniformly across the entire population, without preferential targeting by age, risk group, or co-morbidity. Our core analysis assumed fixed values of 80% efficacy and 80% coverage [27,28]. This resulted in a reduction in TB incidence across the population, applied as a scalar to all incident TB cases (i.e., incidence × [1−efficacy × coverage], equivalent to a 64% reduction in our base case). This reduction then flowed through the model structure to reduce TB deaths, YLL, YLD, and DALYs, with all other epidemiological parameters held constant between intervention and comparator scenarios. We did not include disease transmission dynamics or indirect (herd) protection effects. This simplification allowed us to isolate the impact of analytic choices—specifically cost perspective, timeframe, and post-TB morbidity—on cost-effectiveness results across settings.

**Health system costs.**  Health system costs were assumed unchanged between evaluation approaches; while unspecified these would include the programme cost of the preventative intervention, net of the reduction in cost for future treatment and management of prevented TB cases (assumed up to the point of treatment completion). Scenario analyses separately explored the impact of additional health system costs in the post-TB period (see section "Post-TB health system costs").

**Patient costs.**  Data from twenty countries that completed a National Tuberculosis Patient Cost Survey between 2015 and 2021 have been compiled by the WHO [3]. Costs are provided in USD for the care-seeking and treatment period, separately for first-line and MDR-TB treatment. We used these patient costs for the periods pre-treatment completion, adjusted from the year of the study to 2022 USD values using the Consumer Price Index (CPI) provided by the World Bank (see Table 1). Due to significant hyperinflation since the 2018 survey, we excluded Zimbabwe from analyses. We applied patient costs only to treated patients DS-TB and MDR-TB separately. For Fiji and the Solomon Islands, where only overall treatment costs were available, we applied these costs to all TB patients regardless of resistance status. GDP per capita estimates for 2022 were sourced from the World Bank, and correlation with patient-costs checked using Pearson correlation analysis. No additional patient-level cost data were sourced beyond the WHO's publicly available surveys; all values used are drawn directly from this dataset and adjusted as described. These are presented in Table 1, and Table D in S1 Text, Supporting Information.

Patient cost values for the post-TB period under the extended societal approach were derived from lost income (similar to the 'output approach' [5]). We extracted average household income before and during TB from the patient cost survey data and assumed a linear return to pre-TB income levels following treatment completion. This simplifying assumption was informed by cohort studies from Malawi, India, Brazil and South Africa, which found

**Table 1. Patient Costs and GDP per Capita.**

| Country* | Year of study | GDP per Capita $(2022) | Inflation Factor† | Overall Patient Costs (2022 USD) | | First-line (FL) Patient Costs (2022 USD) | | MDR-TB Patient Costs (2022 USD) | | Ratio of GDP per Capita to Cost | | |
|---|---|---|---|---|---|---|---|---|---|---|---|---|
| | | | | Mean | Range | Mean | Range | Mean | Range | Overall | FL | MDR |
| Brazil | 2020 | $9,455 | 1.18 | $678 | [$576, $798] | $619 | [$523, $734] | $1,439 | [$1,058, $1,960] | 13.9 | 15.3 | 6.6 |
| Burkina Faso | 2020 | $832 | 1.18 | $104 | [$82, $132] | $103 | [$81, $130] | $214 | [$115, $400] | 8.0 | 8.1 | 3.9 |
| DRC | 2019 | $680 | 1.66 | $383 | [$305, $479] | $300 | [$242, $371] | $1,160 | [$752, $1,787] | 1.8 | 2.3 | 0.6 |
| Fiji‡ | 2017 | $5,474 | 1.08 | $336 | [$265, $426] | - | - | - | - | 16.3 | - | - |
| Ghana | 2016 | $2,252 | 2.06 | $941 | [$791, $1,122] | $885 | [$733, $1,069] | $1,676 | [$1,155, $2,431] | 2.4 | 2.5 | 1.3 |
| Indonesia | 2020 | $4,798 | 1.06 | $176 | [$143, $217] | $169 | [$137, $210] | $1,107 | [$902, $1,358] | 27.3 | 28.3 | 4.3 |
| Kenya | 2017 | $2,245 | 1.33 | $139 | [$105, $186] | $138 | [$103, $184] | $803 | [$638, $1,011] | 16.1 | 16.3 | 2.8 |
| Laos | 2018 | $2,047 | 1.39 | $992 | [$845, $1,162] | $982 | [$838, $1,151] | $2,314 | [$1,340, $3,994] | 2.1 | 2.1 | 0.9 |
| Mali | 2021 | $847 | 1.10 | $653 | [$545, $783] | $630 | [$529, $750] | $2,015 | [$1,263, $3,214] | 1.3 | 1.3 | 0.4 |
| Mongolia | 2018 | $4,954 | 1.38 | $1,415 | [$1,187, $1,689] | $1,111 | [$900, $1,371] | $2,758 | [$2,321, $3,274] | 3.5 | 4.5 | 1.8 |
| Myanmar | 2016 | $1,228 | 1.55 | $698 | [$602, $808] | $633 | [$561, $715] | $2,756 | [$2,338, $3,252] | 1.8 | 1.9 | 0.4 |
| Nigeria | 2017 | $2,202 | 1.97 | $900 | [$792, $1,026] | $808 | [$713, $916] | $3,163 | [$2,382, $4,200] | 2.4 | 2.7 | 0.7 |
| PNG | 2018 | $2,622 | 1.20 | $65 | [$50, $84] | $62 | [$48, $80] | $644 | [$493, $844] | 40.5 | 42.1 | 4.1 |
| Philippines | 2017 | $3,624 | 1.21 | $310 | [$280, $342] | $301 | [$272, $333] | $2,126 | [$1,815, $2,489] | 11.7 | 12.0 | 1.7 |
| Solomon Islands‡ | 2019 | $2,205 | 1.09 | $730 | [$412, $1,294] | - | - | - | - | 3.0 | - | - |
| Tanzania | 2019 | $1,253 | 1.12 | $173 | [$154, $194] | $169 | [$151, $188] | $428 | [$264, $693] | 7.2 | 7.4 | 2.9 |
| Thailand | 2021 | $7,070 | 1.06 | $426 | [$374, $484] | $416 | [$366, $472] | $2,748 | [$1,450, $5,207] | 16.6 | 17.0 | 2.6 |
| Uganda | 2017 | $1,103 | 1.19 | $223 | [$192, $258] | $198 | [$176, $222] | $5,004 | [$3,777, $6,629] | 4.9 | 5.6 | 0.2 |
| Vietnam | 2016 | $4,087 | 1.19 | $1,056 | [$924, $1,208] | $925 | [$810, $1,056] | $4,707 | [$4,146, $5,344] | 3.9 | 4.4 | 0.9 |

*DRC: Democratic Republic of the Congo; PNG: Papua New Guinea. †All cost values in this table have been adjusted to constant 2022 US dollars using the listed inflation factor. ‡The low number of MDR-TB patients included in surveys conducted in Fiji and the Solomon Islands precluded a specific estimation of MDR-TB costs. Therefore, our analysis uses the mean overall cost estimates for both DS-TB and MDR-TB patients.

that income and employment often remain suppressed up to 12 months post-treatment, with only partial recovery by two to three years [6–8,29]. This return period was assumed to be 3-years in core analyses (varied from 1 to 10 years in sensitivity analyses), with a 3% annual discount rate applied.

Patient costs associated with receiving the preventative intervention were not included in the core analyses but were explored in a scenario analysis (see section "Patient costs of intervention administration").

**Disability weights.** The disability weight for active TB (0.333), and TB-HIV (0.408) were taken from the GBD [30]. In core analyses we utilised sex and HIV-status specific post-TB weights from a primary post-TB cohort in Malawi, split into two periods: 3-years following treatment completion, and then beyond [21]. We assumed no adverse effects from the intervention in our core results, with both weights and serious adverse events (SAEs) explored in scenario analyses (see Sections "Adverse event from vaccination" and "Alternative post-TB weights", and Table 2).

**Discounting.** For health effects and post-TB patient costs we applied a discount rate of 3%, which was varied in sensitivity analyses.

**Cost-effectiveness threshold.** The relative size of estimated ICER values were explored under our different approaches against threshold values of 0.1 to 3 times GDP per capita, though use 1×GDP per capita (2022) in our core analyses.

## Scenario analyses

We conducted four separate scenario analyses to test the impact of varying our assumptions.

**Patient costs of intervention administration.** We considered how applying a 'patient cost' to those receiving the preventative intervention would change the relative size of ICERs. The Societal Extended ICER remained on the threshold, and the Societal Conventional remained constant (as patient costs prior to TB treatment completion included). For each country, we calculated how high patient costs for receiving the intervention would need to be for the ICER under the Health System Perspective to appear more cost-effective than under the Societal Perspective—i.e., when overall net patient costs were positive.

**Alternative post-TB weights.** We re-ran analyses using a uniform disability weight for the post-TB period (remainder of life) of 0.053, as suggested by Quaife *et al.* (2020) [31].

**Post-TB health system costs.** Empirical follow-up studies suggest approximately 16% of pulmonary DS-TB patients continue to experience disability 3-years post-treatment [21], however, there is little evidence on the magnitude of costs for managing these symptoms. We explored the impact on the Enhanced ICER for a wide range of lifetime costs, from $1 to $100 dollars per person post-TB. This range ($1–$100) was selected to reflect a plausible lower- and upper-bound scenario, given the absence of robust data on post-TB health service utilisation. It draws on limited empirical studies suggesting ongoing disability in a proportion of patients, but acknowledges the wide uncertainty in actual treatment costs.

**Adverse event from vaccination.** Our core analysis assumed no adverse events leading to long-term disability from the intervention, meaning $\Delta D_{Tx} = 0$. We present the effect if a proportion of those who received the intervention suffered an adverse event (incidence$_{SAE}$), applicable if the preventative intervention were, for example, a vaccination. We present two-way data tables, varying the incidence of AEFI (adverse event following immunisation) between 0.01% and 1% and the duration between 0.02 years (~1 week) and 0.5 years. We present three tables for weights of 0.1, 0.2, and 0.3.

This is presented for the country with the largest changes as the percentage increase in the Extended ICER under the societal perspective.

**Table 2. Input parameters for TB Intervention Analysis Including Initial Estimates, Sensitivity Ranges, and Sources.**

| Variable | Definition | Initial Estimate | Range | Source |
|---|---|---|---|---|
| **Specified fixed parameters** | | | | |
| $t_{Tx}$ | Duration of DS-TB treatment | 6 months | | Global Tuberculosis Report 2023 [32] |
| $t_{Tx,MDR}$ | Duration of MDR-TB treatment | 9 months | | Global Tuberculosis Report 2023 [32] |
| **Specified parameters (varied in sensitivity analyses)** | | | | |
| *Disability Weights* | | | | |
| $DW_{aTB}$ | Disability weight for active TB disease | 0.333 | [0.224-0.454] | Global Burden of Disease [30] |
| $DW_{aTB-HIV}$ | Disability weight for TB-HIV coinfection | 0.408 | [0.274-0.549] | Global Burden of Disease [30] |
| $Intvn_{efficacy}$ | Assumed efficacy of preventative intervention† | 80% | - | [27] |
| $Intvn_{efficacy}$ | Assumed coverage of intervention† | 80% | - | [28] |
| $T_{lts}$ | Time taken for household finances to return to pre-TB levels | 3 years | 1-10 years | |
| **Country specific parameters** | | | | |
| $C_{pat,ind,aTB}$ | Patient-incurred cost per TB case (care-seeking & treatment) | - | - | Patient cost surveys [3] |
| $C_{pat,ind,lts}$ | Patient long-term financial shock of TB | - | - | Patient cost surveys [3] |
| | Prevalence of TB by age group | - | - | Global Health Data Exchange (GHDx) [25] |
| | TB mortality incidence | - | - | WHO country reports |
| | MDR-TB prevalence among TB | - | - | WHO country reports [24] |
| | Aspirational life expectancies | - | - | UN Life Tables [26] |
| $CFR$ | Case fatality rate | - | - | WHO country reports [24] |
| | GDP per capita | - | - | United Nations |
| **Unspecified parameters assumed unchanged between approaches** | | | | |
| $C_{vax,prog}$ | Total cost of the intervention programme | - | - | - |
| $C_{treat}$ | Cost of treating one case of TB | - | - | - |
| **Parameters relevant to specific scenario analyses** | | | | |
| *Post-TB disability weights‡* | | | | |
| $DW_{ltd,glob}$ | Disability weight for post-TB (all patients) | 0.053 | applied uniformly (scenario) | Global post-TB DALY modelling study. [31] |
| $DW_{mltd,f3y}$ | DW long-term disability first 3 years (HIV-negative men) | 0.037 | - | Post-TB DALY derivation study [21] |
| $DW_{mltd,3y+}$ | DW long-term disability post 3 years (HIV-negative men) | 0.016 | - | Post-TB DALY derivation study [21] |
| $DW_{fltd,f3y}$ | DW long-term disability first 3 years (HIV-negative women) | 0.053 | - | Post-TB DALY derivation study [21] |
| $DW_{fltd,3y+}$ | DW long-term disability post 3 years (HIV-negative women) | 0.049 | - | Post-TB DALY derivation study [21] |
| $DW_{mltd,f3y}$ | DW long-term disability first 3 years (HIV-positive men) | 0.024 | - | Post-TB DALY derivation study [21] |
| $DW_{mltd,3y+}$ | DW long-term disability post 3 years (HIV-positive men) | 0.009 | - | Post-TB DALY derivation study [21] |
| $DW_{fltd,f3y}$ | DW long-term disability first 3 years (HIV-positive women) | 0.026 | - | Post-TB DALY derivation study [21] |
| $DW_{fltd,3y+}$ | DW long-term disability post 3 years (HIV-positive women) | 0.013 | - | Post-TB DALY derivation study [21] |
| $incidence_{AEFI}$ | Rate of adverse event following immunisation | varied | 0.01% - 1% | [33] |
| $DW_{SAE}$ | Weight applied to those with SAE | varied | 0.1, 0.2, 0.3 | Hypothetical range |
| $T_{SAE}$ | Duration of SAE | varied | 0.02 - 0.5 years | Hypothetical range |

†The values assumed for efficacy and coverage are interchangeable: Incidence$_{reduced}$ = Incidence$_{baseline}$ × (1 − (efficacy × coverage)).

‡ Core analyses used sex- and HIV-specific post-TB disability weights; the uniform 0.053 weight was applied to all post-TB cases only in scenario analyses

### Additional one-way sensitivity analyses

In addition to scenario analyses, which assessed discrete alternative assumptions, we conducted one-way sensitivity analyses to evaluate the impact of parameter uncertainty on disability weights, time taken for financial recovery, and efficacy & coverage (Table 2).

### Ethics

Ethical approval was not sought, as this study was a secondary analysis of previously published data.

## Results

### Country profiles

**Economic - patient costs.**   Patient costs between countries vary considerably, particularly when considered against GDP per captia. The ratio of GDP per capita to overall mean patient-costs ranged from 1.8 times (DRC) to 40.5 times (Papua New Guinea). MDR-TB costs are markedly higher across all countries, with MDR-TB patients in Vietnam experiencing costs nearly 4.5 times that of first-line treatment (DS-TB) patients. Pearson correlation found no significant relationship between GDP per capita and either DSTB patient costs ($p = 0.534$) or MDR-TB costs ($p = 0.927$). (See S1 Fig, S2 Fig, Supporting Information, for graphical illustration). We also found no significant relationship between GDP per capita and individual cost components overall: Direct Medical ($p = 0.830$), Direct Non-medical ($p = 0.920$), and Indirect costs ($p = 0.146$) (see Supporting information, S3 Fig).

**Epidemiological.**   Country TB profiles showed considerable heterogeneity. Philippines registered the highest 2022 incidence (638/100,000) and Burkina Faso the lowest at (44/100,000). On average, the HIV co-infection rate among TB patients across countries was approximately just under 10%, ranging from close to zero (Solomon Islands) to one third of cases in Thailand. TB mortality rate was highest in Myanmar among HIV-negative individuals (80/100,000), while Brazil reported the lowest (3.4/100,000). Average CDR was 68%, again ranging widely from a reported 100% in Uganda to 37% in Ghana (Table B in S1 Text, Supporting Information).

### Intervention and health outcomes

The health benefit offered by the intervention varied substantially between countries across YLL, YLD, and DALYs (Table A in S1 Text). Under the conventional period, the average contribution of YLD to DALYs was 9% across countries, ranging from 4% (Ghana) to 26% (Mongolia). This contribution increased to 31% under the Extended timeframe, (from 14% Ghana up to 60% Mongolia). For both timeframes, DALYs were greatest in Myanmar (2,019 Conventional; 2,527 Extended), with fewest in Fiji (117 & 186). On average, the inclusion in calculations of post-TB disability increased DALY estimates by +36% (ranging +12% Burkina Faso and Ghana to +86% Mongolia). Effectiveness of the intervention was expectantly driven by baseline incidence. The largest benefits were seen in Laos and Vietnam (1,583 and 1,357 DALYs averted, respectively). On average, the extended timeframe increased estimates of DALYs averted by 39% (ranging +13% DRC to +101% Mongolia).

Across all countries, excluding post-TB effects increased ICERs by an average of 51% (ranging from +19.3% in Ghana to +108% in Mongolia), while omitting patient-incurred costs had a smaller effect, , increasing ICERs by an average of 14% (from +0.6% in Papua New Guinea up to +54.4% in Mali).

## Costs

Health system costs were calculated through our methodological assumption that the intervention was cost-effective at the point of the threshold under the societal perspective of the extended evaluation (Fig 3), using a threshold of 1*GDP per capita. Accordingly, HS costs varied between countries, from $803,315 (Brazil) to $93,929 (Burkina Faso). Across all countries patient costs made only a small proportion of total costs, under 10% in all but three countries (Brazil, Burkina Faso, Thailand) under the conventional time period (average 6% conventional period, 14% extended period). Moreover, our post-treatment patient cost estimates were for many countries substantial, on average comprising 64% of total patient costs. These ranged widely, from 21.7% of total patient costs in Solomon Islands to 92% in Burkina Faso.

## ICERs

ICERs calculated under the four approaches (Table 3) demonstrate large variation, with the extended timeframe societal perspective ('Enhanced') evaluation providing the lowest ICER, and the conventional timeframe under the health system perspective ('Basic') the largest (see section "Anchoring ICERs"). This difference was largest for Mongolia at +112.7% Across all countries, the timeframe considered had a greater impact on ICERs than choice of perspective. Expressing differences as percentage change from the smallest ICER, that under the 'Enhanced Evaluation' (societal perspective and the Extended timeframe), a health system perspective with an extended timeframe led to ICERs that are on average 14% larger (ranging +54.4% Mali to +0.6% Papua New Guinea), while a societal perspective under the conventional (shorter) timeframe, increased ICERs by an average of over 50% (+108% Mongolia to +19.3% Ghana).

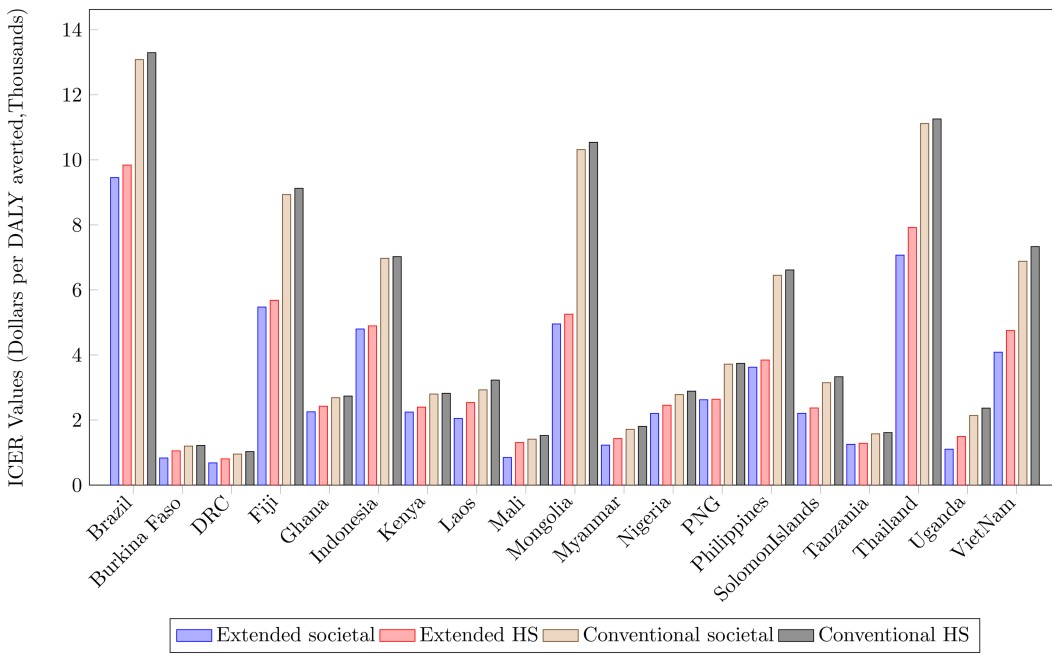

**Fig 3. ICERs for TB Intervention by Country and Approach.** (A standardised figure is provided in Supporting Information, S3 Fig).

## Comparative analysis across thresholds

For all countries, as the size of the cost-effectiveness threshold is increased, the ICER estimations given under the two approaches converge, as the addition of patient costs becomes less relevant to the addition of post-TB DALYs. Fig 4, shows this relationship for each country between threshold values of 0.1×GDP and 2×GDP. Beyond 2×GDP per capita thresholds, the relative size of the ICERs has stabilised, and the difference between the two ICERs for most countries is minimal, although for Mali the Enhanced approach ICER is still 20% lower than the ICER under the conventional approach.

## Sensitivity analyses

Results of sensitivity analyses are summarised in Supporting Information, Tables C,D,E,F,G in S1 Text. Higher and lower values for active TB disability weights (HIV-positive and HIV-negative) affected ICERs in most countries by < ±1%, with the largest difference in Mongolia, where the conventional ICERs (both HS and Soc) increased by 5.6% and 5.8% (decreasing 5.1% and 5.2%). Increasing the period of financial recovery from 3 to 10 years increased costs

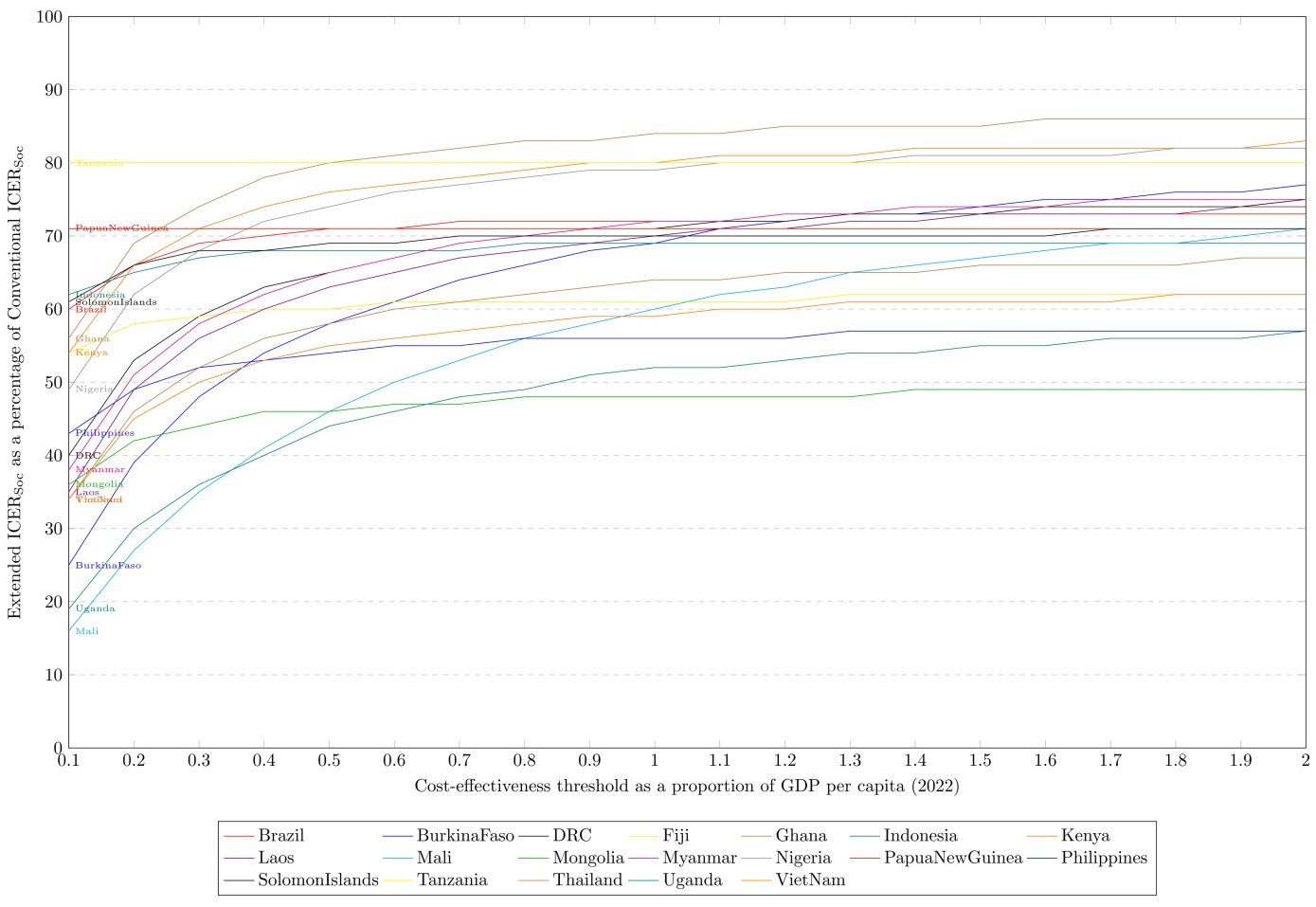

**Fig 4. Relative sizes of ICERs: Societal perspective - Extended vs. Conventional.**

by 202%, while reducing to 1 year decreased costs by 66%. (See Supporting Information Table E in S1 Text); in Papua New Guinea and Tanzania there was no change (see Table 3).

### Scenario analyses

**Including an intervention administration cost to patients.** When assuming that roll-out of intervention incurred patient costs, we found that for even modest costs patient cost savings from preventing treatment were outweighed by the costs of implementation. For the extended evaluation, this ranged from as little as $1 (6 countries) to $6 for Myanmar (Supporting Information, Table F in S1 Text).

**Alternative post-TB weights.** Applying a uniform, lifelong post-TB disability weight of 0.053 increased the Basic ICER relative to the Enhanced by +8% to +43% across countries, with an average increase of +20% (Table I, S1 Text, Supporting Information).

**Adverse events following intervention.** Table J in S1 Text (Supporting Information) shows the percentage increase in the ICER for the intervention under the Extended evaluation, Societal perspective, in the case of Brazil (largest changes). In a sample analysis, for a disability weight of 0.1, for an SAE incidence of 1 in 200, if the duration of SAE lasted 6 months the ICER increased by 16%.

## Discussion

Our analyses demonstrate the significant impact that choices regarding timeframe and analytical perspective can have on cost-effectiveness ratios for a generic preventative TB intervention, such as a vaccine. Across 19 countries, we found that including the post-TB period consistently improved cost-effectiveness estimates, whereas including patient costs had a more limited effect, except at low cost-effectiveness thresholds. The degree of impact varied considerably between countries, reinforcing the importance of context-specific evaluations.

The inclusion of post-TB health effects significantly altered cost-effectiveness ratios. In Mongolia, for example, an intervention that would be considered cost-effective at a threshold of 1×GDP per capita under the extended approach had an ICER twice as large under the conventional approach. The average increase in ICERs across countries was 51%, highlighting the need for greater consideration of post-TB health impacts in cost-effectiveness analyses. While relevant across all intervention types, these impacts are particularly salient for preventative interventions, where TB incidence is averted entirely and with it, all downstream morbidity and disability. The evidence base on prolonged post-TB morbidity is robust [16] and widely recognised by TB care stakeholders [34], yet remains underrepresented in many economic evaluations.

By contrast, including patient costs had a relatively limited effect on ICERs (as estimated using our extended approach). In all countries, their influence diminished as the cost-effectiveness threshold increased, with the relative difference between ICERs (under the societal vs. health system perspectives) plateauing or changing minimally beyond approximately 2×GDP per capita (S5 Fig). At these higher thresholds, patient costs contributed little to the overall ICER. The largest impacts were seen in countries where patient costs were especially high relative to GDP, such as Mali and Uganda, where ICERs were 20% and 11% lower, respectively, under the societal perspective.

**Table 3. ICERs for TB intervention by country and approach.**

| Country | Costs (USD) | | | | DALYs | | ICER | | | |
|---|---|---|---|---|---|---|---|---|---|---|
| | Health‡ System Costs | Patient costs | | | Pre-Tx* completion | Post-Tx completion | Extended timeframe | | Conventional Period | |
| | | Pre-Tx completion | Post-Tx completion | | | | Societal§ Perspective | Health System Perspective | Societal Perspective | Health System Perspective |
| Brazil | 803,315 | 18,190 | 25,924 (58.8%) | | 85.0 | 29.8 | 9,455 | 9,840 (+4.1%) | 13,082 (+38.4%) | 13,296 (+40.6%) |
| Burkina Faso | 93,929 | 2,311 | 26,398 (92%) | | 112.9 | 18.1 | 832 | 1,051 (+26.3%) | 1,199 (+44.1%) | 1,219 (+46.5%) |
| DRC | 477,723 | 53,091 | 59,500 (52.8%) | | 703.0 | 195.7 | 680 | 805 (+18.4%) | 953 (+40.1%) | 1,029 (+51.3%) |
| Fiji | 399,753 | 14,028 | 10,154 (42%) | | 73.0 | 44.3 | 5,474 | 5,680 (+3.8%) | 8,933 (+63.2%) | 9,125 (+66.7%) |
| Ghana | 1,395,678 | 31,099 | 89,936 (74.3%) | | 619.8 | 79.5 | 2,252 | 2,425 (+7.7%) | 2,686 (+19.3%) | 2,736 (+21.5%) |
| Indonesia | 3,204,646 | 33,742 | 60,028 (64%) | | 667.9 | 290.3 | 4,798 | 4,896 (+2%) | 6,973 (+45.3%) | 7,024 (+46.4%) |
| Kenya | 1,656,258 | 17,323 | 114,034 (86.8%) | | 737.7 | 130.8 | 2,245 | 2,397 (+6.8%) | 2,798 (+24.6%) | 2,821 (+25.7%) |
| Laos | 500,338 | 73,644 | 78,493 (51.6%) | | 244.4 | 66.9 | 2,047 | 2,536 (+23.9%) | 2,928 (+43%) | 3,229 (+57.7%) |
| Mali | 115,091 | 15,385 | 57,487 (78.9%) | | 135.8 | 22.4 | 847 | 1,308 (+54.4%) | 1,411 (+66.6%) | 1,524 (+79.9%) |
| Mongolia | 1,499,821 | 68,157 | 115,058 (62.8%) | | 302.8 | 304.4 | 4,954 | 5,255 (+6.1%) | 10,314 (+108.2%) | 10,539 (+112.7%) |
| Myanmar | 1,542,325 | 110,856 | 210,353 (65.5%) | | 1256.3 | 326.7 | 1,228 | 1,431 (+16.5%) | 1,714 (+39.6%) | 1,803 (+46.8%) |
| Nigeria | 1,612,965 | 76,086 | 139,647 (64.7%) | | 732.3 | 129.6 | 2,202 | 2,453 (+11.4%) | 2,783 (+26.4%) | 2,887 (+31.1%) |
| PNG† | 2,508,182 | 21,817 | - | | 956.7 | 400.1 | 2,622 | 2,638 (+0.6%) | 3,718 (+41.8%) | 3,741 (+42.7%) |
| Philippines | 2,115,099 | 94,932 | 127,177 (57.3%) | | 583.7 | 420.4 | 3,624 | 3,845 (+6.1%) | 6,451 (+78%) | 6,614 (+82.5%) |
| Solomon Islands | 269,751 | 22,404 | 6,218 (21.7%) | | 122.4 | 49.5 | 2,205 | 2,371 (+7.5%) | 3,147 (+42.7%) | 3,330 (+51%) |
| Tanzania† | 605,905 | 19,196 | - | | 483.6 | 124.0 | 1,253 | 1,284 (+2.5%) | 1,574 (+25.6%) | 1,614 (+28.8%) |
| Thailand | 1,551,987 | 30,929 | 235,012 (88.4%) | | 219.5 | 92.4 | 7,070 | 7,922 (+12.1%) | 11,116 (+57.2%) | 11,257 (+59.2%) |
| Uganda | 193,540 | 39,633 | 68,869 (63.5%) | | 175.4 | 102.3 | 1,103 | 1,494 (+35.4%) | 2,139 (+93.9%) | 2,365 (+114.4%) |
| Vietnam | 719,585 | 79,466 | 101,201 (56%) | | 176.1 | 95.7 | 4,087 | 4,751 (+16.2%) | 6,883 (+68.4%) | 7,334 (+79.4%) |

*All DALYs due to mortality are included under pre-treatment completion (Pre-Tx completion), with post-treatment DALYs only including estimate of post-TB morbidity. (Further breakdown of YLD in Supporting Information, Table A in S1 Text). † Cost data for Papua New Guinea (PNG) and Tanzania suggest household income was higher during TB treatment than prior [3], so the calculation for post-TB financial recovery was not possible and assumed to be zero. ‡Total HS costs (including all DS-TB and MDR-TB) were estimated indirectly through the cost-effectiveness threshold, assuming the extended approach under the societal perspective is cost-effective at 1×GDP per capita. §Percentages in brackets in the final columns are relative to the Extended ICER under the societal perspective.

## Cost-effectiveness thresholds and GDP influence

Cost-effectiveness thresholds (often determined as multiples of GDP per capita) play a critical role in how much patient-incurred costs affect the ICER. Because our ICER calculations were anchored to GDP, the relative influence of adding patient costs depended on this. In absolute terms, patient-incurred costs constitute a larger share of total TB costs in low-GDP countries, making the difference between including or excluding those costs appear larger. In higher-GDP settings, the same absolute patient costs represent a smaller fraction of the threshold, so their inclusion produces a smaller change in ICER. For example, in Mali (GDP per capita ≈ $847), adding patient costs lowered the ICER by about 20%, whereas in Brazil (GDP per capita ≈ $10,000), the ICER difference was only  4% at a 1×GDP per capita threshold.

Notably, we found no significant overall correlation between a country's GDP and the magnitude of TB patient costs, yet several countries with the highest average patient cost burdens (e.g., Laos, Ghana, Nigeria) also have relatively low GDP per capita, likely due to weaker health financing arrangements and social protection systems leading to higher out-of-pocket expenses. This highlights an important issue, being that analyses using GDP-based thresholds may place less weight on patient costs in wealthier settings, potentially obscuring the true financial burden of TB, especially in middle-income countries where patient costs remain high but appear less influential proportionally. When we examined results in relative terms (percent change in ICER), patient-incurred costs were still a substantial component across all settings, suggesting that GDP-based ICER thresholds can mask the impact of these costs rather than eliminate it. Grouping countries by World Bank income classification, we found that excluding patient-incurred costs increased ICERs by an average of 12.6% in low-income, 11.3% in lower-middle-income, and 5.9% in upper-middle-income countries, supporting the observation that the relative influence of patient-incurred costs on cost-effectiveness estimates diminishes in higher-income settings with larger GDP-based thresholds.

There is ongoing debate about the appropriateness of GDP-based cost-effectiveness thresholds. Critics have noted their limitations and proposed alternative approaches, such as benchmarking against context-specific interventions or using "league tables" [35]. Despite these critiques, GDP-based thresholds remain common in practice [36]. Additionally, GDP-based thresholds cannot necessarily account for the pricing structure of some interventions, such as vaccines, which may be subject to tiered pricing or global procurement mechanisms that may not align closely with national income levels.

As a result, cost-effectiveness assessed using GDP-based thresholds may undervalue such interventions in lower-income settings, given that intervention costs remain relatively consistent across countries whereas affordability differs. Our findings support the adoption of more nuanced, equitable, and context-specific criteria for cost-effectiveness—ones that better reflect each country's economic realities and protect against undervaluing interventions in lower-income settings. Incorporating measures of financial risk protection or affordability into threshold-setting (in addition to or instead of pure GDP-per-capita metrics) could enhance cross-country comparisons and decision-making fairness.

## Societal perspective and externalities

The application of a "societal" perspective in TB cost-effectiveness studies is often more restricted than the term suggests. The relatively low impact of patient costs on ICERs in our results should not be misinterpreted as meaning those costs can be ignored or that adopting a societal perspective is unimportant. Rather, it reflects the fact that the patient costs typically included in TB cost-effectiveness studies (as in our "conventional" analysis) capture only a narrow part of true societal costs. Even our extended scenario with a longer post-treatment

economic recovery period would be more accurately described as a "limited societal" perspective, as defined by Kim et al. [11].

While consensus on definitions is lacking, generally, a 'societal perspective' is expected to consider the public interest with wider spillover effects beyond the health sector, often encompassing factors such as productivity losses and cost-shifting between sectors, and potentially too the effects on sectors such as education [37,38]. Such considerations are often difficult to fully capture in practice, with patient cost estimates typically constrained by data availability and methodological choices. Moreover, reliance on the DALY as the primary health outcome measure—focusing on individual health losses rather than wider economic and social disruptions—further distances such analyses from a truly comprehensive societal perspective. As a result, even CEAs adopting a societal perspective are overlooking key externalities, particularly those related to household and social wellbeing—highly relevant in the context of TB. These challenges become even more pronounced in preventative interventions, where public health externalities are clearer, and specific frameworks have been proposed to address them [39,40].

Our study has several limitations that must be considered when interpreting the findings. First, we did not account for the effects of onward TB transmission, which would likely increase the cost-effectiveness of preventative interventions by reducing secondary cases. This simplification was intentional: by excluding dynamic transmission and indirect protection effects, we aimed to maintain consistency across countries and isolate the impact of key analytic choices in a transparent and replicable manner. Additionally, we assumed no differences in TB case detection or case fatality rates with or without the intervention, a likely simplification of real-world dynamics. Moreover, several of the WHO-derived parameters—such as TB incidence, case fatality rates, and detection rates—have considerable uncertainty, often provided with wide confidence intervals. While our model is instrumental, this uncertainty could influence the scale of our findings, particularly for country-level estimates, and should be borne in mind. Second, our analysis was constrained by a lack of long-term data on both patient and health system post-TB costs. While we assumed that household finances recover approximately three years after TB treatment completion, this is based on limited empirical evidence and likely varies by setting [6–8,29]. The recovery was modelled as linear, which may not reflect actual income trajectories in all contexts. Similarly, post-TB healthcare costs remain poorly documented, and our analysis does not capture the financial burden associated with managing long-term TB-related complications—costs which could meaningfully influence cost-effectiveness estimates if systematically included. Given this uncertainty, we applied a wide ($1–$100) range in sensitivity analyses to explore plausible scenarios.

Although our population was modelled to reflect TB/HIV co-infection rates using TB-HIV specific disability weights, we did not explicitly model differences in outcomes by HIV status, which could impact cost-effectiveness if interventions affect these groups differently. Furthermore, our analysis assumes TB care is delivered within government- or donor-financed programmes, where patients face minimal direct treatment costs. In settings with insurance-based financing or cost-sharing mechanisms, patient costs could be higher, potentially altering ICERs. Lastly, while we presented results across a range of cost-effectiveness thresholds, the use of GDP-based thresholds remains a subject of debate, as they may not fully capture societal willingness-to-pay or affordability considerations. Future research incorporating alternative threshold-setting approaches and detailed financial protection metrics could provide a more comprehensive view of TB intervention cost-effectiveness.

### Policy implications and future research

A key motivation for this work is to inform TB policy and resource allocation by providing evidence on long-term outcomes. A 2022 qualitative study highlighted significant data gaps that hinder informed decision-making by funders and policy-makers, noting that many National TB Programmes feel unable to advocate for post-TB care due to a lack of funding, guidelines, and evidence. Our study helps address this evidence gap by demonstrating the significant impact of the post-TB period on cost-effectiveness and, by extension, on policy decisions. For example, a TB vaccine that appears marginally cost-effective under a conventional analysis may actually be highly cost-effective when post-TB disability and patient costs are included—potentially influencing financing decisions in low-income countries.

By quantifying how much extending the analytic horizon beyond the final day of medication can change cost-effectiveness results, we provide policymakers with an estimate of the value of reconceptualising TB as a disease with outcomes that extend into the post-treatment period. Our results advocate for longer-term follow-up in economic evaluations of TB interventions to ensure that benefits such as restored productivity, averted disability, and prevented long-term complications are captured.

### Conclusion

In conclusion, our findings underscore that narrowly defining the scope of TB cost-effectiveness analyses can lead to an incomplete picture of an intervention's value. Analyses that consider only the costs and health effects up to treatment completion (and that focus solely on health system costs) risk missing a substantial portion of TB's total impact. We observed that when the enduring health and economic consequences of TB—such as post-treatment disability and ongoing patient financial burdens—are included, preventative interventions appear considerably more cost-effective.

Perspectives that include these broader costs and longer-term outcomes increase the likelihood that an intervention will be deemed cost-effective, especially in lower-income settings or at lower willingness-to-pay thresholds. Conversely, failing to account for TB's extended impacts under-represents the benefits of preventative measures, which could bias decisions against interventions that are actually highly valuable in the long run. Ultimately, incorporating post-TB outcomes and patient costs into cost-effectiveness evaluations will provide a more accurate reflection of the true benefits of TB interventions, helping to guide better-informed and more equitable policy decisions.

Our findings suggest that traditional cost-effectiveness analyses may systematically underestimate the value of TB prevention. Researchers should consider adopting longer timeframes and broader cost perspectives when evaluating interventions to ensure that their full societal and economic benefits are recognised.

## Supporting information

**S1 Text. Tables and additional Supporting information.**
(PDF)

**S1 Fig. DS-TB Patient costs in relation to country GDP per Capita.**
(EPS)

**S2 Fig. MDR-TB Patient costs in relation to country GDP per Capita.**
(EPS)

**S3 Fig. Breakdown of patient cost components ('Overall') in relation to GDP per capita.**
No significant correlation was found between GDP per capita and either total patient cost or
individual cost components (Pearson correlation, p > 0.05 for all).
(EPS)

**S4 Fig. ICERs for TB Intervention by Country and Approach normalised to percentage
change from Extended Societal ICER.**
(EPS)

**S5 Fig. Relative sizes of ICERs against different threshold values.**
(EPS)

**S6 Fig. Relative sizes of ICERs against different threshold values.**
(EPS)

## Acknowledgments

The authors express their gratitude to Dr. Rein Houben, Dr. Alice Zwerling, and Dr. Imelda
Bates for their insightful feedback and suggestions, which strengthened the final version of
this manuscript.

## Author contributions

**Conceptualization:** Ewan M. Tomeny, S. Bertel Squire, Eve Worrall.

**Data curation:** Ewan M. Tomeny, Laura Rosu.

**Formal analysis:** Ewan M. Tomeny.

**Methodology:** Ewan M. Tomeny, Joseph Kazibwe, Georgios F. Nikolaidis, Eve Worrall.

**Project administration:** Ewan M. Tomeny, Eve Worrall.

**Resources:** Ewan M. Tomeny.

**Software:** Ewan M. Tomeny, Georgios F. Nikolaidis.

**Supervision:** Rebecca Nightingale, Tom Wingfield, Jamilah Meghji, S. Bertel Squire, Eve
Worrall.

**Validation:** Ewan M. Tomeny, Phuong Bich Tran, Joseph Kazibwe.

**Visualization:** Ewan M. Tomeny, Eve Worrall.

**Writing – original draft:** Ewan M. Tomeny.

**Writing – review & editing:** Ewan M. Tomeny, Phuong Bich Tran, Joseph Kazibwe, Laura
Rosu, Rebecca Nightingale, Tom Wingfield, Jamilah Meghji, S. Bertel Squire, Eve Worrall.

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
