## [Decision Letter · Decision Letter 0]

6 May 2025

PGPH-D-25-00584

A broader lens on TB cost-effectiveness analysis: How patient-incurred costs and post-TB outcomes reshape estimates in a multi-country study

Dear Dr. Tomeny,

Thank you for submitting your manuscript to PLOS Global Public Health. After careful consideration, we feel that it has merit but does not fully meet PLOS Global Public Health’s publication criteria as it currently stands. Therefore, we invite you to submit a revised version of the manuscript that addresses the points raised during the review process.

We look forward to receiving your revised manuscript.

Kind regards,

Angela Devine, PhD

Academic Editor

Journal Requirements:

Additional Editor Comments (if provided):

Reviewers' comments:

Reviewer's Responses to Questions

**Comments to the Author**

1. Does this manuscript meet PLOS Global Public Health’s publication criteria? Is the manuscript technically sound, and do the data support the conclusions? The manuscript must describe methodologically and ethically rigorous research with conclusions that are appropriately drawn based on the data presented.

Reviewer #1: Yes

Reviewer #2: Yes

2. Has the statistical analysis been performed appropriately and rigorously?

Reviewer #1: Yes

Reviewer #2: Yes

3. Have the authors made all data underlying the findings in their manuscript fully available (please refer to the Data Availability Statement at the start of the manuscript PDF file)?

Reviewer #1: Yes

Reviewer #2: Yes

4. Is the manuscript presented in an intelligible fashion and written in standard English?

Reviewer #1: Yes

Reviewer #2: Yes

5. Review Comments to the Author

Reviewer #1: This is an excellent, and highly important, paper. My congratulations to the authors who have developed a very nice way of demonstrating and articulating the importance of extending CEA timeframes and perspectives. The paper is very well-written overall, and I only have a few comments - most of which are minor.

Comments

1. You don’t currently explain how you estimate health benefit from vaccines in any detail. Can you expand section 2.3.4 to explain in further detail how vaccines were modelled, including any targeting or transmission impacts.

2. In the equations on page 5 and 6, it might be useful to distinguish costs incurred by the health system during the TB episode and costs incurred in the post-TB period. This would make the equations match your Figure 1.

3. “Assuming the sex- and age distribution within each country was unchanged from 2019, we took 5-year breakdown from the IHME Global Health Data Exchange.” – I don’t quite understand this sentence. 5-year breakdown of what?

4. It would be useful to be more specific about your source of cost data – it doesn’t look like the WHO report cited provides data specific enough for you to have included in your analysis? Did you receive the actual data from WHO?

5. I realise that this is just an instrumental model, so properly managing uncertainty may not be necessary, but it would be useful to mention this in your methods/discussion. Some of the WHO estimates have very wide error bars – how would this uncertainty affect your findings?

6. Do you have anything to justify your assumption of a linear return to pre-TB income, and the time frame for return? Also, In the text you say the range is 1-10 years, and in Table 2 this is listed as 0-10 years.

7. In Table 1, can you label whether the cost data are presented in 2022 USD or in the year of study

8. Section 3.1.1 – it might be worth testing correlation of direct and indirect costs separately.

Reviewer #2: The paper is well written and addresses important concepts of perspectives and timeframes adopted when conducting cost-effective analysis. The authors make a valuable contribution by investigating their impact on TB ICERs. Their results justify using more detailed approaches in the economic evaluations of TB.

- A limitation of the study is the exclusion of the healthcare costs associated with long-term disability. It is justified in figure 1’s footnote, but it is an important point and should be mentioned in the methods section, and its implications on the results should be more expanded on in the text.

- In 4.1, paragraph 2, p.22, there is a very good point made about the weight on patient costs in wealthier settings and the country’s GDP. Did you try grouping the countries by income level and checking how the results compare?

- More details on the costs included in the patients incurred cost would help understand what is taking into account and put the results into perspective.

Minor comments:

1. In the title, it might be better to avoid using abbreviations and write tuberculosis instead of TB.

2. In the introduction, p.3, “The societal perspective, rooted in welfare economics, [12] aims to maximise gains for society as a whole, incorporating all relevant costs and consequences, irrespective of who incurs them. While theoretically broad, its real-world application is often constrained, though the inclusion of patient costs is expected.” I find the last sentence a little bit confusing. Maybe if we add a sentence to explain more why it is constrained and a link to the last part of the sentence.

3. Some abbreviations are in capital letters and some are not (ds-TB in the last paragraph p.12, dalys last paragraph p.18), it would be good to be consistent. Also, SAE is not defined the first time it is used in the manuscript.

4. Last paragraph p.12, lifetime costs were varied between $1 and $100. It would be good to mention what informed this range.

5. Last sentence p.16: the average effect of excluding post-TB effects on ICERs is quantified, but that of omitting patient-incurred costs is not here. It might be good to have similar numbers (average and range) for this part as well.

6. Discussion: the discussion is well structured, and very interesting points were addressed.

a. In the 2nd paragraph, “The average increase in ICERs across countries was 51%, highlighting the need for greater consideration of post-TB health impacts in cost-effectiveness analyses, particularly for preventative interventions.”, can you elaborate on why it is particularly for preventative interventions.

b. In the 3rd paragraph, the cost-effectiveness threshold is 2*GDP. It might be good to add a sentence and explain why it is different from the 1*GDP in the 2nd paragraph.

c. Point 4.1, 3rd paragraph: “Additionally, GDP-based thresholds cannot fully account for the pricing structure of interventions such as vaccines, which are often globally standardised rather than varying proportionally with GDP”, the second part of the sentence is not very accurate as vaccine prices are not globally standardised. BCG is very cheap across countries, but the sentence seems broader than BCG.

6. PLOS authors have the option to publish the peer review history of their article (what does this mean?). If published, this will include your full peer review and any attached files.

**Do you want your identity to be public for this peer review?** For information about this choice, including consent withdrawal, please see our Privacy Policy.

Reviewer #1: **Yes: **Sedona Sweeney

Reviewer #2: No

---

## [Decision Letter · Decision Letter 1]

31 Jul 2025

A broader lens on tuberculosis cost-effectiveness analysis: How patient-incurred costs and post-tuberculosis outcomes reshape estimates in a multi-country study

PGPH-D-25-00584R1

Dear Tomeny,

We are pleased to inform you that your manuscript 'A broader lens on tuberculosis cost-effectiveness analysis: How patient-incurred costs and post-tuberculosis outcomes reshape estimates in a multi-country study' has been provisionally accepted for publication in PLOS Global Public Health.

Best regards,

Angela Devine, PhD

Academic Editor

Reviewer Comments (if any, and for reference):

Reviewer's Responses to Questions

**Comments to the Author**

1. If the authors have adequately addressed your comments raised in a previous round of review and you feel that this manuscript is now acceptable for publication, you may indicate that here to bypass the “Comments to the Author” section, enter your conflict of interest statement in the “Confidential to Editor” section, and submit your "Accept" recommendation.

Reviewer #2: All comments have been addressed

2. Does this manuscript meet PLOS Global Public Health’s publication criteria? Is the manuscript technically sound, and do the data support the conclusions? The manuscript must describe methodologically and ethically rigorous research with conclusions that are appropriately drawn based on the data presented.

Reviewer #2: Yes

3. Has the statistical analysis been performed appropriately and rigorously?

Reviewer #2: Yes

4. Have the authors made all data underlying the findings in their manuscript fully available (please refer to the Data Availability Statement at the start of the manuscript PDF file)?

Reviewer #2: Yes

5. Is the manuscript presented in an intelligible fashion and written in standard English?

Reviewer #2: Yes

6. Review Comments to the Author

Reviewer #2: The authors have thoroughly addressed the points raised. The manuscript reads very well and delivers an important and clear message.

I only have one editing comment which is to remove the " from the end of the second paragraph of section 4.1

7. PLOS authors have the option to publish the peer review history of their article (what does this mean?). If published, this will include your full peer review and any attached files.

**Do you want your identity to be public for this peer review?** For information about this choice, including consent withdrawal, please see our Privacy Policy.

Reviewer #2: No
